# The Influence of Medicine Approaches on Patient Trust, Satisfaction, and Loyalty

**DOI:** 10.3390/healthcare11091254

**Published:** 2023-04-27

**Authors:** Ayşe Sertan, Kemal Çek, Adile Öniz, Murat Özgören

**Affiliations:** 1Faculty of Health Sciences, Near East University, via Mersin 10, 99138 Nicosia, Turkey; 2Faculty of Economics and Administrative Sciences, Accounting and Finance, Cyprus International University, via Mersin 10, 99138 Nicosia, Turkey; 3Healthcare Organizations Management, Dean of Faculty of Health Sciences, Near East University, via Mersin 10, 99138 Nicosia, Turkey; 4Faculty of Medicine, Near East University, via Mersin 10, 99138 Nicosia, Turkey

**Keywords:** traditional and complementary medicine approach, effectiveness of the treatment, patient trust, patient satisfaction, patient loyalty

## Abstract

The increase in traditional and complementary medicine (TCM) methods has revealed the necessity of determining relevant behavioral tendencies among healthcare users. In our study, the evaluation of TCM approaches of healthcare users living in North Cyprus in terms of treatment effectiveness, patient trust and patient satisfaction, and the effects of these variables on patient loyalty, were investigated. Data were collected utilizing the scale approach as well as the survey technique in order to measure the variables in the study. A total of 452 participants completed the survey. TCM has a positive effect on treatment effectiveness, patient trust and patient satisfaction. Patient trust and patient satisfaction have a positive effect on patient loyalty, whereas treatment effectiveness did not have a significant effect on patient loyalty. There is a significant and positive mediating effect of patient trust and patient satisfaction between TCM and patient loyalty. However, the mediating effect of treatment effectiveness is not significant between TCM and patient loyalty. This study will help researchers and practitioners understand the importance of attitude, trust, effectiveness, satisfaction and loyalty in relation to TCM. It is suggested that studies that measure the behaviors of patients should increase in order to obtain better health outcomes.

## 1. Introduction

It is essential to ensure that all people and communities have access to the healthcare they want and need, without financial strain. Traditional medicine can now be considered a kind of basic healthcare for people. Traditional medicine has taken a position in many people’s daily health-seeking habits, particularly in the cultural area, and has become an important component of health services [1]. Traditional medicine is the collection of information, skill, and practices that can or cannot be explained based on cultural beliefs and experiences, and which are employed in the maintenance of health, as well as the prevention of physical and mental disorders [2]. Traditional medicine can also be characterized as beliefs and practice systems that emerge independently in different cultures over time [3]. Definition of symptoms, methods and criteria of diagnosis, treatments and evaluating their results are a part of this system and all medical systems are part of a theoretical framework that exists within the depths of cultural context. Symptom definitions, diagnostic procedures and criteria, therapies and outcome evaluation are all intrinsic in these systems, and all medical systems operate within a theoretical framework that is deeply embedded within a specific cultural context. Complementary medicine, on the other hand, is an application that is thought to provide an additional advantage to persons as a result of the employment of health practices that are in accordance with medicine [4]. It has been noticed that complementary medicine has a supportive structure that, in addition to the use of current medical techniques, provides a soothing impact on the patient, boosts the patient’s immunity and improves the patient’s psychological condition [5]. However, there has recently been a major growth in the number of TCM practitioners [6].

TCM applications, which are continually evolving, have begun to spread globally and are retaining their popularity. People have started to research, prefer and use TCM all over the world. One of the important reasons for the development of TCM applications is the fact that the health industry has become a major business, i.e., it has essentially become a market on its own. Individuals cannot benefit equally from the offered health services; thus, it leads to individuals looking for different alternatives and being interested in health solutions that may be easier to access [7]. Another health-related challenge is accessing the extremely costly drugs that stand out as a health problem for individuals in many countries and strain the economies of individuals. Access to medicines is difficult for individuals in many parts of the world. This situation leads societies to develop and trust traditional treatments, which are formed within their own cultural structure. Especially in industrialized countries, the naturally forged demand to use different alternative methods in order to reduce health expenditures or to battle chronic diseases with natural methods causes TCM applications to become widespread [8]. TCM is applied with many methods, such as acupuncture, ozone therapy, homeopathy, mesotherapy, oxygen therapy, massage, ayurveda, aromatherapy, hypnosis, cryotherapy, yoga, osteopathy, reflexology, spa therapy, thermal therapy, SPA therapy, hydrotherapy, music therapy and Pilates. These practices can be transmitted from generation to generation, especially based on past experiences and observations, and they can also be specific to the country in which they are applied [2]. Due to the biological diversity of Cyprus, plants native to Cyprus can be used in medicine and the pharmaceutical industry [9]. According to WHO’s TCM treatments report (2019), herbal medicines are mostly used in Cyprus and the legal status of herbal medicines is the same as the traditional medicine regulation published in 2005. Periodic inspections are carried out by the authorities at the production facilities to ensure compliance. Herbal medicine and homeopathy providers have been nationally regulated since 2006. TCM practitioners work in the private sector and the National Government issues the required TCM license to practice [2].

A patient who has a positive perception of treatment can perceive the treatment’s effectiveness as positive. Positively perceived treatment can enhance trust and satisfaction and thus, result in loyalty.

The current study adds to the literature in certain ways: It is one of the first known studies to investigate patients and TCM. For this reason, it reveals that studies that measure the behaviors of patients should increase in order to obtain better health outcomes. The success of TCM applications is due to the effective, efficient and safe application of these methods by professionals with expertise in this field, and the high benefits to be created for individuals in this direction. As a result, in order to assess the size of the impact of TCM applications, which are rapidly spreading among individuals, it is necessary to determine the degree of satisfaction with these applications, how much trust they have in them, the effectiveness of the treatments provided and the loyalty of individuals towards the applications. Following that, showing the relationship between the proper connections and these progressive variables will allow us to see the development direction of these practices, which take on a new dimension in the eyes of people as a whole. Academic and administrative research on TCM applications in the field of health has increased evaluation studies on this subject in recent years. The purpose of this study is to assess the TCM perspectives of people living in North Cyprus in relation to treatment effectiveness, patient trust, and patient satisfaction, as well as the effects of these variables on patient loyalty.

## 2. Research Model and Hypothesis

According to the present literature, individuals can go to whichever health institution they want based on their economic status and health insurance, and this condition has caused people’s expectations to differ and change. The demand for TCM applications, which has been spreading rapidly, especially in recent years, has created a competitive environment in the health sector. It is clear that all health institutions value the effectiveness of the health service or treatment they provide, as well as patient happiness, trust, and loyalty, and that they endeavor to improve the quality of the health services supplied. Therefore, concepts such as service quality, patient attitude, and patient loyalty in health services are areas worth researching.

The concept of attitude can be explained in three different contexts based on how it is formed and how it changes in individuals. First, although it may occur in connection with the values, general goals, language, emotions and development of individuals, the person’s attitude may change at this point. The second context is the social relations that affect the attitudes of individuals, especially with the influence of social media and culture. The third context is socio-historical and affects the attitudes of individuals to socio-political, economic and climatic events. Therefore, attitude can shape or change people’s attitudes in personal, social and historical contexts [10]. A person’s attitude influences their confidence in a service or product. Trust, in particular, appears to be a common issue, faced by every society. In the health sector, this issue becomes more complicated. In particular, the communication skills and clinical abilities of healthcare providers can affect their ability to form trust in patients [11]. Therefore, trust is of critical importance. In this context, developing effective and transparent communication, mutual respect and offering trust-based communication to forecast and solve difficulties can promote mutual trust building [12]. Based on the literature, the study put forward the following hypothesis:

**H1.** 
*Traditional and complementary medicine attitudes are positively associated with patient trust.*


The extent of the relationship between physician and patient varies. This has become a challenging process for healthcare providers. Trust and communication seem to play a vital role in a healthy and effective doctor-patient relationship. Individuals can better adapt to the therapy offered and the medical advice given if trust and communication are formed, according to the papers analyzed in connection with the subject, and these factors are components of success in all fields of study [13]. As a result, ensuring healthy communication between health professionals and patients in TCM practices will favorably change individuals’ TCM attitudes, and at the same time, it will cause the patient’s trust in these practices to advance in a favorable manner.

Satisfaction can be measured by how well the user experience, as a result of utilizing the product or service, satisfies the buyer’s expectations regarding the value [14]. There is an expectation that buyers perceive before purchasing and experiencing products or services. In this setting, firms must produce a surprising effect by going above and beyond their anticipated expectations in order to meet buyers’ requirements and desires [15]. Patient satisfaction is an increasing problem in all aspects of healthcare, particularly in the healthcare industry. It is critical to understand how patients are treated and whether the care they receive satisfies their expectations. In this context, a strong patient-provider relationship should be maintained. Patient satisfaction is frequently expressed in terms of effectiveness and efficiency, particularly in the literature [16]. Based on the literature, the study put forward the following hypothesis:

**H2.** 
*Traditional and complementary medicine attitude is positively associated with patient satisfaction.*


Patient satisfaction is considered an indication of health service quality. Patient satisfaction can be defined as the attitude and compliance of practitioners and patients, loyalty, clinical results, technical competence, accessibility and efficacy within the boundaries of the healthcare offered [17]. From this perspective, educating the patients by providing them with health-related information, improving critical knowledge about diseases, and developing self-health management and self-efficacy will all be crucial practices to strengthen the patient. As a result, in order to effect a behavioral change in individuals and achieve quality patient-care outcomes, healthcare practitioners must collaborate with patients in the process of educating the patients in order to give enough information and boost patient satisfaction [18]. Effective communication between healthcare practitioners and patients, in particular, has been found to have a high level of satisfaction-enhancing effects on patient experiences [19]. As a result, the effectiveness of the treatments used by health professionals who are also involved in TCM applications, the positive clinical results, the optimal level of competence and accessibility, the necessary level of patient awareness, and effective communication will all have a positive impact on individuals’ TCM attitudes. At the same time, this will result in an increase in patient satisfaction with these practices.

Efficiency is evaluated as achieving the desired result for defined resources [20]. Individuals’ treatment responses and symptoms may be instructive for treatment practitioners in the health industry [21]. To begin with, for the treatment to be effective, there must be sufficient pre-clinical justification and proof, as well as applications that are dependable, especially for long-term use in clinical practice. Accordingly, data from high-quality, coordinated clinical research conducted in many locations throughout the world are required [22]. Based on the literature, the study put forward the following hypothesis:

**H3.** 
*Traditional and complementary medicine attitude is positively associated with the effectiveness of the treatment.*


In order to determine treatment efficacy, comparisons of active treatments should be made, and patients, clinicians and interventions representing the practice should be examined [23]. For an effective intervention to be effective in clinical practice, it must be available, service providers must identify its target population, recommend the intervention, and health users must accept and be loyal to the intervention. Individuals with insufficient access, advice, acceptance, and compliance rates may render high-impact interventions less effective than less effective interventions during the implementation phase [24]. As a result, while making healthcare decisions, it is critical to consider the views of patients and service providers during the implementation process. In particular, in order for the treatment to be effective and produce positive benefits for the individual; individuals and service providers must comply, accept and be loyal to the treatment, in addition to the treatment’s high degree of efficacy.

An individual’s attitude can be shown by the information they have in the face of an event, case, or circumstance, as well as their favorable or negative feelings toward that occurrence, as well as their discourses and actions on the subject [25]. Knowing individuals’ attitudes, in particular, will aid in predicting how they will behave in the face of an event and taking action to mitigate the repercussions that will generate unfavorable attitudes. This practice can only be achieved by measuring attitudes in a reliable way. In order to measure attitudes, specially developed measurement techniques and methods should be considered [26]. Accordingly, the scales that can predict the holistic TCM attitude that will affect the level of trust, satisfaction and efficacy of TCM practices in individuals were utilized in this study.

Patients’ trust in healthcare professionals is at the heart of clinical practice. Patients, in particular, must trust their doctors with their lives and health, and doctors must act in such a way that this confidence is maintained. Other health professionals, such as nurses and psychotherapists, are subject to similar requirements under the code of conduct. The foundation of effective treatments and patient-centered care in this setting is directly proportional to patients’ faith in healthcare providers [27]. One of the most significant aspects of the treatment and recovery process is the effective relationship between the physician and the patient. The physician must create friendly communication with the patient and must learn and practice this ability. This talent indicates that the physician can readily collect the required knowledge about their patient’s physical and mental states, and as a result, they will be able to determine the most appropriate approach to treatment for the patient. In particular, the treatment process of patients is affected by the trust of the doctor and the communication between them, rather than the recommended drugs [28]. Loyalty, on the other hand, is a behavior that encourages the performance of institutions and efforts are made to gain it. The repurchase or non-purchase of specific services, the search for diversity and the buyer’s presence or absence of preferences are all elements that influence loyalty [29]. In this regard, patient loyalty can be characterized as a patient liking the service provided by a health institution, preferring the same health institution again based on their needs, promoting it to others and embracing the health institution [30]. Patient trust produced by successful communication between physician and patient, patient satisfaction with effective treatment and positive results as a result of efficient communication all have a favorable impact on patient loyalty [31]. In this context, it is evident that linear and intermediary interactions influence an individual’s attitude, trust in the healthcare they receive, satisfaction and effectiveness of treatment, and patient loyalty made at the end of the process. Based on the literature, the study put forward the following hypothesis:

**H4a.** 
*Patient trust mediates the relationship between traditional and complementary medicine attitudes and patient loyalty.*


**H4b.** 
*Patient satisfaction mediates the relationship between traditional and complementary medicine attitudes and patient loyalty.*


**H4c.** 
*Treatment effectiveness mediates the relationship between traditional and complementary medicine attitudes and patient loyalty.*


In light of the previous literature, the following research model is presented in model 1. Figure 1 below shows the research model of the study.

## 3. Materials and Methods

### 3.1. Design

The study is a cross-sectional type of research. It is a quantitative study in which numerical data were obtained using descriptive and relational research models, and within these data, the current status of the subject was investigated using the descriptive method and whether there was a relationship between the relational method and the variables.

### 3.2. Participants

Data were collected simultaneously both face-to-face and online between January and March 2022 due to COVID-19 restrictions. The online survey is distributed through social media channels to randomly selected individuals from North Cyprus. This study includes all individuals over the age of 18 participants, randomly selected from the patient population living in North Cyprus using the disproportionate stratified sampling method. Because it would be difficult to contact the whole population of the research in terms of time, expense and examination, the convenience sampling approach was used to select a sample to reflect the study population in the research. Therefore, the random sampling procedure obtained a representative sample of the target group. In Structural Equation Modelling methods, the sample size should be large enough (*n* > 200) to reduce the sampling error. Therefore, a random sample was used. Sekeran and Bougie (2013) suggested that 384 would be a sufficient sample size for a population above 100,000 [32]. A total of 452 participants from various regions of North Cyprus participated in the survey.

### 3.3. Variables

Data were collected utilizing the scale approach as well as the survey technique in order to measure the variables in the study. The study employed a sociodemographic questionnaire form, a TCM attitude scale, a treatment efficacy scale, a patient confidence scale, a patient satisfaction scale and a patient loyalty scale. The validity and reliability of the scales have been approved and are open to access.

The Sociodemographic Questionnaire Form prepared by the researchers consists of questions regarding gender, age, education level, occupation/work status, region of residence, which TCM methods are used and frequency of use. The TCM Attitude Scale is a 5-point Likert rating scale consisting of 27 items [33,34], originally developed by Mc Fadden et al., while Köse et al. approved its Turkish validity and reliability. The Treatment Efficacy Scale for Medication is the Abbreviated Treatment Satisfaction Questionnaire (TSQM). It consists of 4 scales: efficacy scale, side effects, fitness, and overall satisfaction scale. Fourteen questions were determined in line with the information obtained from the literature review and focus groups. The four scales of the TSQM include the efficacy scale (questions 1 to 3), the side effects scale (questions 4 to 8), the relevance scale (questions 9 to 11) and the overall satisfaction scale (questions 12 to 14) [35]. The Public/Private Healthcare Provider Trust Scale developed by Ozawa was used to measure patient trust. The scale consisting of 10 statements is a 5-point Likert scale [36,37]. In order to measure patient satisfaction, questions that measure patient satisfaction, consisting of 9 statements, whose validity and reliability was approved by Yesilyurt, will be used. It is a 5-point Likert scale [38]. For this study, the Servqual Scale was taken as a base; it was developed and adapted. The patient Loyalty Scale, which is 11 statements in this questionnaire, measures patient loyalty in one aspect. Its validity and reliability have been established and it is a 5-point Likert Scale [39].

### 3.4. Data Analysis

Statistical analyses were evaluated using IBM AMOS V21. (Analysis of Moment Structures). While evaluating the data, descriptive statistical methods (number, percentage, mean, standard deviation) were used. According to the distribution characteristics of the data, parametric and non-parametric tests and statistics determining the relationship were used.

## 4. Results

This study includes 452 participants that live in Cyprus and represent the population from the different regions of the country, and that are evaluated statistically to take part in the research. The majority of participants are female: 239 participants (52.9%). The majority of participants are university-educated or higher: 288 participants (63.7%). The majority of participants are from Nicosia: 130 participants (28.8%). The demographic characteristics of the participants and the distribution of the regions they live in are given in Table 1 below.

The distribution of the participants in terms of awareness about the term TCM, usage status and frequency of use is given in Table 2 below.

The majority of participants in the research have heard of the term TCM: 322 participants (71.2%). The majority of participants use both traditional and complementary medicine: 282 participants (62.4%). Most of the participants use it very rarely (39.9%).

In the attitudes of the participants towards traditional and complementary and modern treatments, the effectiveness of treatment effectiveness, patient trust, patient satisfaction and patient loyalty were examined. Patient trust, treatment effectiveness and patient loyalty variables show clearer expressions in individuals. In patient satisfaction, expectations and satisfaction conditions differ and it is obvious that the effects vary among individuals. In this direction, the attitudes of the participants towards traditional and complementary treatments. This has led to the preference of modern medicine in terms of trust, effectiveness and loyalty. Since there are more relative conditions in patient satisfaction, traditional and complementary treatments and modern treatments have been determined at a close level. Efficiency distribution by attitude is given in Figure 2 below.

Confirmatory factor analysis (CFA) has been conducted to investigate the factor loadings and the validity of the constructs of the study. Table 3 below represents the results of the CFA. As a cut-off criteria, a 0.5 threshold has been used. Furthermore, the model fit is tested by using the following indices: the comparative fit index (CFI), the goodness of fit index (GFI), the chi-square mean/degree of freedom (CMIN/df), the root means square error (RMSEA) and the standardized root mean square residual (SRMR). According to Hair et al. [40], the suggested threshold points for a good model fit should have CFI and TLI above 0.90, RMSEA below 0.05 and SRMR below 0.09. The model fit is found to be acceptable as suggested by the fit indices (CMIN/df = 1.974, *p* < 0.05, comparative fit index (CFI) = 0.97, goodness of fit index (GFI) = 0.91, root mean square error of approximation (RMSEA) = 0.04 and standardized root mean square residual (SRMR) = 0.04). To test the discriminant validity of the study, AVE and the squared correlation between the variables have been checked. In addition, the method of single latent variable has been applied to test the common method variance. The results have implied that common method variance does not exist in this study.

Table 4 below shows the reliability and validity of the constructs. According to Hair et al. [40], AVE should be above 0.50 threshold in order to achieve convergent validity; CR should be above 0.70 threshold in order to achieve reliability and MSV should be below AVE to achieve discriminant validity of the constructs. The results are shown in Table 4 below. Thus, the findings indicated that the validity and reliability of the constructs are acceptable.

The relationship between the participants’ perspectives and their attitudes toward TCM, treatment effectiveness, patient trust, patient satisfaction and patient loyalty were investigated and the results are presented in Table 5 below. There was no significant association between the treatment efficacy and the patient loyalty, although there were significant relationships between the mean of the other categories (*p* < 0.05). It is found that TCM has a significant and positive effect on the treatment efficacy, patient trust and patient satisfaction. These results confirm H1, H2 and H3. In addition, the results showed that patient trust and patient satisfaction had a positive and significant effect on patient loyalty.

The participants’ perspectives on the relationship between treatment effectiveness, patient trust and satisfaction, TCM attitude, and patient loyalty were investigated. The mediation and indirect effect of treatment effectiveness, patient trust and satisfaction were investigated by employing 95% bias-corrected bootstrapped confidence intervals (N = 5000). There was no significant mediation impact between treatment efficacy, TCM attitude and patient loyalty; however, there was a significant and positive mediating effect in other criteria (*p* < 0.05). These results confirm H4a and H4b but do not confirm H4c. The representation of the results is given in Table 6 below.

## 5. Discussion

With the diversification of individual wants and needs, it is clear that the search for new orientations in the health system, particularly TCM treatment approaches, is on the agenda, and that they are fast growing and popular all around the world. This is an indication that the healthcare market is changing and that there will be increasingly fierce market competition among healthcare institutions. Therefore, providing satisfactory healthcare to patients, gaining the trust of patients and, as a result, acquiring loyal patients are key in the competitive process between healthcare institutions. After reviewing the literature, this study is one of the few studies in which these relations with TCM attitude are made. The study aimed to evaluate the TCM approaches of healthcare users living in Northern Cyprus with treatment effectiveness, patient trust and patient satisfaction, and to investigate the effects of these variables on patient loyalty. This study showed that attitude towards TCM is affected by patient trust, patient satisfaction, and patient loyalty. For example, trust and satisfaction are directly related to each other. Patients who trust their doctor will be more likely to be satisfied with the service and care provided. With this positive feedback loop, ensuring trust will result in better service and increase trust and happiness [41]. In addition, the trust established between the doctor and the patient will have a favorable impact on the patient’s dedication to healthcare [42]. High levels of trust and patient satisfaction indicate a positive patient-provider relationship [43]. However, in line with the favorable results, the individual’s confidence, satisfaction, and effectiveness with the treatment will also have a positive effect on patient loyalty. As a result, addressing the factors that influence patient loyalty will be critical to healthcare practitioners’ financial success [44]. Ng et al. (2022) found that attitude towards TCM was the most influential predictor of intention to use TCM, followed by satisfaction and subjective norms. A full mediating effect was found on the relationship between attitude, knowledge and intention to use TCM, while satisfaction was found to have a full mediating effect on the relationship between perceived service quality and intention to use TCM [45]. In addition, this study showed that patient satisfaction was positively related to the attitude towards TCM. Li et al. (2021) analyzed the effect of the service quality of TCM treatments on patient loyalty and the mediating effect of patient attitude. It was found that the perceived service quality of TCM had a positive effect on patient loyalty and this relationship was mediated by patient attitude [46]. In another study by Ng et al. (2021), a positive relationship was found towards the use of TCM treatments, the level of attitude, and satisfaction [47]. Our study also confirms that there is a positive relationship between attitude and satisfaction towards TCM. Cao et al. (2020) found that treatment effectiveness, participation in TCM treatments and the attitudes of practitioners are the factors that determine participation and compliance with treatment. It is stated that the trust and expectation of users towards TCM treatment has a facilitating effect on compliance [48]. In this study, it was concluded that there was no significant relationship in the mediating effect of treatment efficacy between TCM attitude and patient loyalty. More studies are needed to examine treatment efficacy between TCM attitude and patient loyalty. In order for us to discuss the effectiveness of TCM treatments, it is necessary to base them on a scientific basis and to mention more evidence-based and generally accepted benefits. It is suggested to promote education, training and regulations for both practitioners and the public and to raise public awareness. Our findings also provided empirical support for the mediating role of attitude towards TCM in patient behavior. High levels of trust and satisfaction directly increased TCM attitude, but also increased patient loyalty. The effect of trust, satisfaction and loyalty to TCM is explained by attitude. Contrary to expectations, treatment efficacy did not have a direct significant effect on patient loyalty. These results are in line with the current study, which shows that the TCM attitude in individuals has direct and mediating positive effects on patient trust and patient satisfaction. The fact that individuals have a positive experience in the use of TCM treatment methods and recommend these methods to other individuals with this experience reveals that confidence and satisfaction can increase and their attitudes towards TCM applications can change positively.

Examining the behavioral dimension of individuals against TCM treatments may facilitate the introduction of TCM. In this study, traditional and complementary medicine and patient behaviors are examined. Patient trust, satisfaction, effectiveness and loyalty are considered. The treatment effect does not mediate. Trust and satisfaction mediate results. The findings can assist key stakeholders, including TCM practitioners, consumers and policy makers, to improve the process to improve positive attitude towards TCM treatments and improve service quality. Evidence-based TCM treatments need to be developed to increase the use of TCM treatments in Northern Cyprus. At this point, public practices and support are important. The present study provides theoretical contributions to researchers. To the best of our knowledge, this study is the first to investigate patient behavior related to TCM treatments in Northern Cyprus. The findings of this study have an improving effect on how the researchers’ behavior towards TCM treatments is also affected by different factors. The results may provide insight for future research for researchers to explore different views on patient behavior in TCM treatments in Northern Cyprus.

The study has certain limitations: It was performed in a single country and it was prepared with the current opinions of the participants, as a one-time analysis. In order to increase comparability, the study can be conducted in different countries. Finally, higher number of participants could not have been reached due to COVID-19 restrictions.

## 6. Conclusions

In conclusion, this study makes a significant contribution to the literature on individuals receiving TCM care. In the results, significant relationships were found between the direct effects and mediating effects of patient trust and patient satisfaction on patient loyalty. This research helps researchers and practitioners to understand the importance of attitude, trust, effectiveness, satisfaction, and loyalty to the advantages and disadvantages of treatments at the point where they evaluate the behavior of individuals towards traditional and complementary medicine methods. Providing community awareness training with public support and integrating these practices into primary healthcare will help them lay the foundation for improving health-related outcomes. It is suggested that studies that measure the behaviors of patients should increase in order to obtain better health outcomes.

A supportive approach by all stakeholder groups in the health sector would ultimately lead to a more efficient, safer, and more effective use of TCM treatment practices.

## Figures and Tables

**Figure 1 healthcare-11-01254-f001:**
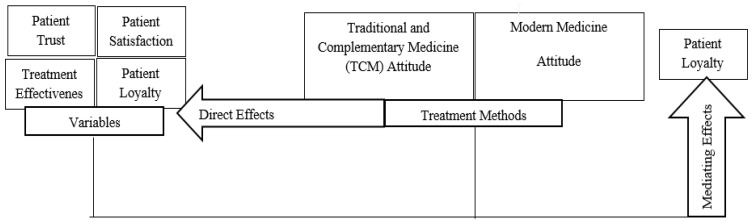
Research model. The circles are labeled as the main research parameters in this study. The central section denotes the two major attitudes against treatment methods. The flow of conceptual relationships is indicated by arrows.

**Figure 2 healthcare-11-01254-f002:**
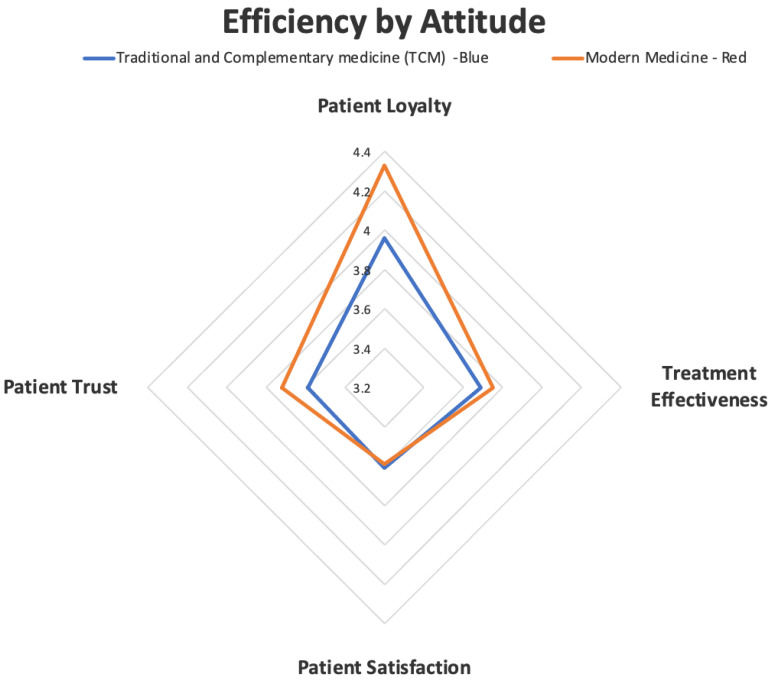
Efficiency by attitude as shown by a radar plot. The four dimensions are displayed for both traditional and complementary medicine TCM (blue) and modern medicine (red) scores as patient loyalty, patient trust, treatment effectiveness and patient satisfaction. The magnitude is depicted in 0–5 Likert Scale.

**Table 1 healthcare-11-01254-t001:** Demographic characteristics.

		Number of Participants (N)	%
Gender	Female	239	52.9
	Male	213	47.1
	Total	452	100.0
Education	Primary School	9	2.0
	Secondary School	35	7.7
	High School	119	26.3
	University or Higher	288	63.7
	Total	452	100.0
Region	Lefke	21	4.6
	İskele	28	6.2
	Güzelyurt	72	15.9
	Famagusta	98	21.7
	Kyrenia	103	22.8
	Nicosia	130	28.8
	Total	452	100.0

**Table 2 healthcare-11-01254-t002:** Awareness of traditional and complementary medicine, usage status and frequency of use by participants.

		Number of Participants (N)	%
Awareness	Yes	322	71.2
	No	130	28.8
	Total	452	100.0
Usage Status	Traditional Medicine	112	24.8
	Complementary Medicine	58	12.8
	Traditional and Complementary Medicine (Both)	282	62.4
	Total	452	100.0
Frequency of Use	Extremely Rarely		39.9
	Rarely		37.0
	Usually		15.9
	Often		4.6
	Every time		2.6
	Total		100.0

**Table 3 healthcare-11-01254-t003:** Factor loadings.

Constructs	Items	Factor Loadings	Significance
TCM	TCM1	0.665	***
	TCM2	0.629	***
	TCM3	0.665	***
	TCM4	0.776	***
	TCM5	0.740	***
	TCM6	0.644	***
	TCM7	0.631	***
EFF	EFF1	0.759	***
	EFF2	0.900	***
	EFF3	0.807	***
TRST	TRST1	0.742	***
	TRST2	0.771	***
	TRST3	0.574	***
	TRST4	0.633	***
	TRST5	0.783	***
	TRST6	0.742	***
SAT	SAT1	0.743	***
	SAT2	0.668	***
	SAT3	0.593	***
	SAT4	0.751	***
	SAT5	0.820	***
	SAT6	0.828	***
LYL	LYL1	0.800	***
	LYL2	0.696	***
	LYL3	0.755	***
	LYL4	0.774	***
	LYL5	0.811	***
	LYL6	0.729	***
	LYL7	0.867	***
	LYL8	0.786	***

Notes: *** Significant; TCM, traditional and complementary medicine; EFF, treatment effectiveness; TRST, patient trust; SAT, patient satisfaction; LYL, patient loyalty.

**Table 4 healthcare-11-01254-t004:** Reliability and validity of the constructs.

	CR	AVE	MSV	MaxR(H)	SAT	TCM	EFF	TRST	LYL
SAT	0.875	0.585	0.336	0.884	0.765				
TCM	0.857	0.663	0.301	0.864	0.543	0.810			
EFF	0.863	0.679	0.058	0.882	0.240	0.219	0.824		
TRST	0.853	0.539	0.359	0.859	0.529	0.549	0.126	0.734	
LYL	0.925	0.607	0.359	0.930	0.580	0.541	0.180	0.599	0.779

Notes: CR, composite reliability; AVE, average variance extracted; MSV, maximum shared variance; MaxR(H), maximum reliability; SAT, patient satisfaction; TCM, traditional and complementary medicine; EFF, treatment effectiveness; TRST, patient trust; LYL, patient loyalty.

**Table 5 healthcare-11-01254-t005:** Direct effects between variables.

Parameter			Estimate	Lower	Upper	P	Results
EFF	<---	TCM	0.408	0.301	0.504	0.010	*** H3 Supported
TRST	<---	TCM	0.860	0.768	0.931	0.010	*** H1 Supported
SAT	<---	TCM	0.846	0.761	0.902	0.010	*** H2 Supported
LYL	<---	TRST	0.196	0.054	0.339	0.049	***
LYL	<---	EFF	−0.041	−0.098	0.027	0.245	-
LYL	<---	SAT	0.837	0.726	0.951	0.010	***

Notes: *** Significant; - Insignificant; TCM, traditional and complementary medicine; EFF, treatment effectiveness; TRST, patient trust; SAT, patient satisfaction; LYL, patient loyalty.

**Table 6 healthcare-11-01254-t006:** Mediating effects between variables.

Parameter	Estimate	Lower	Upper	P	Results
TCM --> EFF --> LYL	−0.017	−0.044	0.010	0.245	H4c Not Supported
TCM --> TRST --> LYL	0.174	0.047	0.278	0.049	*** H4a Supported
TCM --> SAT --> LYL	0.734	0.562	0.901	0.010	*** H4b Supported

Notes: *** Significant; TCM, traditional and complementary medicine; EFF, treatment effectiveness; TRST, patient trust; SAT, patient satisfaction; LYL, patient loyalty.

## Data Availability

The data presented in this study are available on request from the corresponding author.

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
