# Peer review of "The Influence of Medicine Approaches on Patient Trust, Satisfaction, and Loyalty"

_healthcare, 2023, doi:10.3390/healthcare11091254_

Round 1
Reviewer 1 Report
Overall, the manuscript is good. Following questions need to be addressed before further consideration.
The result section of the abstract needs complete revision.
Introduction
Describe in detail the use of different types of TCM in Cyprus.
What is the legal status of TCM in Cyprus?
Methods
Figure 1. the red mark should be removed.
It is not easy to follow figure 1. Please revise it and make it more understandable.
Why both online and face-to-face methods were employed?
What method of dissemination was used for online data collection?
How have you used random and non-random sampling? Please clarify.
How the sample size was calculated?
How the tool was validated?
Indicate the AMOS version and name of the company.
Results
I would suggest authors, not to start the first sentence with digits like;
“52.9% of the individuals participating in the research are female”
Discussion
It is currently inadequate and needs serious efforts to update backed by the latest articles.
Information regarding the limitations should be added.
Conclusions
Please revise it and describe only the key conclusive statements of your study.
Author Response
Reviewer 1:
Overall, the manuscript is good. Following questions need to be addressed before further consideration.
Reviewer’s Comment
The result section of the abstract needs complete revision.
Author’s Response:
The abstract has been updated in accordance with the comments.
Reviewer’s Comment
Introduction
Describe in detail the use of different types of TCM in Cyprus.
What is the legal status of TCM in Cyprus?
Author’s Response:
TCM in Cyprus is explained in detail.
Reviewer’s Comment
Methods
Figure 1. the red mark should be removed.
It is not easy to follow figure 1. Please revise it and make it more understandable.
Why both online and face-to-face methods were employed?
What method of dissemination was used for online data collection?
How have you used random and non-random sampling? Please clarify.
How the sample size was calculated?
How the tool was validated?
Indicate the AMOS version and name of the company.
Author’s Response:
Fig. 1 marks is removed and fixed.
Due to covid-19 restrictions both methods were employed to be able to reach the desired sample size.
Sample size is clarified.
AMOS version and company is mentioned.
Reviewer’s Comment
Results
I would suggest authors, not to start the first sentence with digits like;
“52.9% of the individuals participating in the research are female”
Author’s Response:
The results section is fixed.
Reviewer’s Comment
Discussion
It is currently inadequate and needs serious efforts to update backed by the latest articles.
Information regarding the limitations should be added.
Author’s Response:
Discussion have been improved with more recent articles.
Limitation have been added.
Reviewer’s Comment
Conclusions
Please revise it and describe only the key conclusive statements of your study.
Author’s Response:
Conclusion is revised accordingly.
"Please see the attachment."

Reviewer 2 Report
The purpose of this study is to assess the TCM perspectives of people living in North Cyprus in relation to treatment effectiveness, patient trust, and patient satisfaction, as well as the effects of these variables on patient loyalty.
The authors describe in detail the methodology of their project, applying a multidimensional approach to the analysis of the issue. However, they did not sufficiently emphasize the originality of their research.
The obtained test results confirm the data published in the literature.It is true that the authors point out that their results can be used by medics, but still no indication of their novelty.
No description of study limitations.
Conclusions could include specific guidelines for medical practice.
Author Response
Reviewer 2:
The purpose of this study is to assess the TCM perspectives of people living in North Cyprus in relation to treatment effectiveness, patient trust, and patient satisfaction, as well as the effects of these variables on patient loyalty.
The authors describe in detail the methodology of their project, applying a multidimensional approach to the analysis of the issue. However, they did not sufficiently emphasize the originality of their research.
The obtained test results confirm the data published in the literature.It is true that the authors point out that their results can be used by medics, but still no indication of their novelty.
No description of study limitations.
Conclusions could include specific guidelines for medical practice.
Author’s Response:
1-Originality of the research is mentioned in the discussion section.
2-Limitation is added to the discussion section.
3-Guidelines for medical practice is added to the conclusion section.
"Please see the attachment."

Round 2
Reviewer 1 Report
No further comments.